# Analysis of Intra-Tumoral Macrophages and T Cells in Non-Small Cell Lung Cancer (NSCLC) Indicates a Role for Immune Checkpoint and CD200-CD200R Interactions

**DOI:** 10.3390/cancers13081788

**Published:** 2021-04-09

**Authors:** Anders Tøndell, Yashwanth Subbannayya, Sissel Gyrid Freim Wahl, Arnar Flatberg, Sveinung Sørhaug, Magne Børset, Markus Haug

**Affiliations:** 1Department of Thoracic Medicine, St. Olavs Hospital, Trondheim University Hospital, 7006 Trondheim, Norway; Sveinung.Sorhaug@stolav.no; 2Department of Clinical and Molecular Medicine, Norwegian University of Science and Technology, 7491 Trondheim, Norway; yashwanth.subbannayya@ntnu.no (Y.S.); sissel.g.f.wahl@ntnu.no (S.G.F.W.) arnar.flatberg@ntnu.no (A.F.); magne.borset@ntnu.no (M.B.); markus.haug@ntnu.no (M.H.); 3Centre of Molecular Inflammation Research, Norwegian University of Science and Technology, 7491 Trondheim, Norway; 4Department of Pathology, St. Olavs Hospital, Trondheim University Hospital, 7006 Trondheim, Norway; 5Central Administration, St. Olavs Hospital, Trondheim University Hospital, 7006 Trondheim, Norway; 6Department of Immunology and Transfusion Medicine, St. Olavs Hospital, Trondheim University Hospital, 7006 Trondheim, Norway; 7Department of Infectious Diseases, St. Olavs Hospital, Trondheim University Hospital, 7006 Trondheim, Norway

**Keywords:** non-small cell lung cancer, immunosuppression, tumor microenvironment, tumor-associated macrophages

## Abstract

**Simple Summary:**

Lung cancer is the leading cause of cancer-related death worldwide, accounting for nearly one-fifth of all cancer-related deaths. Immunotherapy with immune checkpoint inhibitors has become one of the most promising approaches in the treatment of advanced lung cancer, although beneficial responses are seen only in a proportion of patients. To improve immunotherapy treatment responses in lung cancer, we need to identify which immunosuppression mechanisms are activated in the tumor microenvironment. In this study, we investigated gene expression profiles in intra-tumoral immune cells in lung cancer, focusing on tumor-associated macrophages, and interactions with CD4^+^ and CD8^+^ T cells. Our data highlight two newly described immunosuppressive pathways, which may represent novel innate immune checkpoints dampening the anti-tumor T cell immune response in lung cancer. Our results substantiate the importance of tumor-associated macrophages as a mediator of immunosuppression and a promising target for immunotherapy.

**Abstract:**

Non-small cell lung carcinoma (NSCLC) is one of the most commonly diagnosed cancers and a leading cause of cancer-related deaths. Immunotherapy with immune checkpoint inhibitors shows beneficial responses, but only in a proportion of patients. To improve immunotherapy in NSCLC, we need to map the immune checkpoints that contribute immunosuppression in NSCLC-associated immune cells and to identify novel pathways that regulate immunosuppression. Here, we investigated the gene expression profiles of intra-tumoral immune cells isolated from NSCLC patients and compared them to the expression profiles of their counterparts in adjacent healthy tissue. Transcriptome analysis was performed on macrophages, CD4^+^ and CD8^+^ T cells. The data was subjected to Gene Ontology (GO) term enrichment and weighted correlation network analysis in order to identify mediators of immunosuppression in the tumor microenvironment in NSCLC. Immune cells from NSCLC revealed a consistent differential expression of genes involved in interactions between myeloid cells and lymphocytes. We further identified several immunosuppressive molecules and pathways that may be activated in tumor-associated macrophages in NSCLC. Importantly, we report novel data on immune cell expression of the newly described CD200/CD200R1 pathway, and the leukocyte immunoglobulin-like receptors (LILRs), which may represent novel innate immune checkpoints, dampening the anti-tumor T cell immune response in NSCLC. Our study substantiates the importance of tumor-associated macrophages as a mediator of immunosuppression and a promising target for immunotherapy.

## 1. Introduction

Lung cancer is the most commonly diagnosed cancer, and the leading cause of cancer-related death [1], accounting for an estimated 1.8 million deaths worldwide in 2018, which is nearly one-fifth of all cancer-related deaths [2]. Non-small cell lung carcinoma (NSCLC) accounts for about 85% of all lung cancers and has a predicted five-year survival rate of 23% [2]. One reason for this low survival rate is that the majority of patients are diagnosed with advanced disease.

Immunotherapy with immune checkpoint inhibitors targeting the programmed cell death protein 1 (PD-1) or the programmed cell death protein ligand 1 (PD-L1) has become one of the most promising approaches in the treatment of advanced NSCLC [3,4]. However, clinical studies show beneficial long-term responses only in a proportion of patients [5,6] and effective immunotherapy options are still lacking for the majority of patients with advanced NSCLC. This may be caused by innate or acquired resistance to immune checkpoint inhibitors [7], e.g., due to a compensatory upregulation of other checkpoint receptors [8]. In addition, the accuracy of biomarkers, such as immunohistochemical analysis of PD-L1 expression in tumor cells, seems not sufficient to guide the stratification of patients for immunotherapy [4]. In the NSCLC tumor microenvironment, several mechanisms for immunosuppression may operate in concert [9] as a combination of immunotherapy targeting both the PD-1/PD-L1 axis and cytotoxic T lymphocyte protein 4 (CTLA-4) in cancer patients has given promising results [10]. Immunotherapies attacking several novel targets are currently in clinical trials or are under development [11].

Tumor-associated macrophages (TAMs) are capable of inducing tumor-promoting signals in the tumor microenvironment (TME) [12], and increased frequencies of TAMs are associated with poor prognosis [13]. In general, TAMs exert an inhibitory effect on the anti-tumor immune response and engage in bidirectional interactions with other cells, such as CD4^+^ and CD8^+^ T cells [13]. Innate immune checkpoints related to phagocytosis may play a key role in tumor-mediated immune escape, and may potentially become targets for cancer immunotherapy [14]. In a previous study, our group reported higher expression of CD39 on T cells in the TME of NSCLC than on T cells derived from healthy adjacent tissue [15]. CD39 and CD73 on T cells, macrophages, and tumor cells hydrolyze ATP to AMP and adenosine. In turn, adenosine deliver immunosuppressive signals to immune cells such as macrophages and T cells via adenosine receptors. These events constitute another important mechanism for immunosuppression involving TAMs in the TME [16,17]. Another group of inhibitory innate immune receptors expressed on TAMs and T cells, the leukocyte immunoglobulin-like receptors (LILRs), have been proposed as immune checkpoints with a tumor-supportive role [18]. Targeting of innate immune checkpoints on TAMs may, therefore, provide a promising novel therapeutic approach [19]. Current knowledge of the activation status of immunosuppressive mechanisms and immune checkpoints in intra-tumoral immune cells in NSCLC, and, in particular, the contribution from TAMs, is still too unsatisfactory to allow stratification of patients for different combination immunotherapy regimens.

Here, we investigated the gene expression profiles of intra-tumoral macrophages, CD4^+^ and CD8^+^ T cells from NSCLC patients, and compared them to the expression profiles of their counterparts in healthy adjacent tissue. The objective was to identify specific markers and expression patterns that might affect function of these intra-tumoral immune cells and to identify potential new targets for future immunotherapy. We focused on possible cross-talk between receptor/ligand pairs on intra-tumoral macrophages and T cells, which might modify the effector functions of intra-tumoral immune cells. Several immune checkpoint receptor/ligand pairs were dysregulated in intra-tumoral immune cells, including the CD200-CD200R interaction pathway between T cells and macrophages.

## 2. Materials and Methods

In the time period between February 2017 and March 2018, 11 patients receiving lung or lobe resection for NSCLC treatment were included in the present study. All patients had subsolid or solid tumors with a minimum size of 20 mm in diameter as estimated by a thoracic CT scan. None of the patients had a previous history of lung cancer and all were treatment-naïve. Histological classification and subtyping were performed according to the 2015 WHO classification of lung tumors [20]. The study population included seven male and four female patients with a median age of 73 years (range 56–83). Five patients had squamous cell carcinomas, and six had adenocarcinomas. All study participants provided written informed consent and the study was approved by the Regional Committee for Medical and Health Research Ethics (REC Central, Ref.nr.: 2010/1939). A summary of the main characteristics of the study population is outlined in Table 1.

Tumor and adjacent normal lung tissue were obtained by video-assisted thoracic surgery or open thoracic surgery in patients with NSCLC. Within 30 min after surgery, tissue from tumor and macroscopically normal lung (volume range of 1–3 cm^3^) was obtained from the lung lobes. Mononuclear immune cells were extracted by a combined enzymatic and mechanical disaggregation protocol [21], as detailed in our previous study [15]. Tumor tissue disintegration and immune cell extraction are illustrated in Appendix A. The procedure included rinsing with serum-free Hyclone DMEM/F12 media (Thermo Fisher Scientific, Waltham, MA, USA), followed by slicing of the samples into 1–2 mm^3^ pieces by sterile microdissection, and incubation at 37 °C in Hyclone Leibovitz L-15 media (Thermo Fisher Scientific) supplemented with the enzymes collagenases I (170 mg/L), II (56 mg/L), IV (170 mg/L), DNase I (25 mg/L), and elastase (25 mg/L) (all Worthington Biochemical, New Jersey, US). Samples were disintegrated on a shaker (85 rpm, 45 + 30−50 min, interrupted by pipetting for 5–10 min) and filtered through a 70 µm cell strainer (BD Biosciences, San Jose, CA, USA). Mononuclear cells were further isolated by subsequent density gradient centrifugation (Lymphoprep, Stemcell technologies Cambridge, UK) and lysis of remaining red blood cells with BD Pharm Lyse buffer (BD Bioscience). The mononuclear cells were washed twice and resuspended in cell culture medium (Hyclone DMEM/F12 supplemented with 10% fetal calf serum (FCS), Thermo Fisher Scientific).

For surface antigen staining, the mononuclear cells were labeled with eFluor780-Fixable Viability dye (eBioscience, San Diego, CA, USA) and incubated for 15 min with fluorescence-labeled monoclonal antibodies to CD206 (Brilliant Violet 421), CD45 (Brilliant Violet 510,), CD8 (PerCP/Cyanine5.5), and CD4 (PE-CF594) (all from BD Biosciences, San Jose, CA, USA). The cells were washed and resuspended in PBS with 2% FCS in sterile low-adhesion polypropylene tubes (Corning, New York, NY, USA) and sorted on a BD FACS Aria III cell sorter (BD Biosciences) with BD FACSDiva 8.0 software (BD Biosciences). The FACS gating strategy and post-sort validation of cell purity is exemplified in Appendix A. Viable CD45^+^ Side Scatter (SSc)^low^ and Forward Scatter (FSc)^low^ cells were identified as lymphocytes. CD4^+^ and CD8^+^ T cells were identified as CD4^+^ or CD8^+^ cells in the lymphocyte gate. Macrophages were identified as viable high autofluorescence (empty FITC channel^high^) CD45^bright^ SSc^high^ CD206^+^ cells. In tissue-resident macrophages, such as in a normal human lung, CD206 is ubiquitously expressed in macrophage populations [22,23], and in the dominating M2-polarized population of TAMs in NSCLC [24]. We did not include CD14 in the gating strategy, as this might have excluded immunosuppressive myeloid cells with low expression of CD14 [25,26]. Sorted immune cells were collected in Roswell Park Memorial Institute medium 1640 (RPMI, Sigma-Aldrich Corp, St. Louis, MO, USA) supplemented with 30% FCS in sterile polypropylene tubes (Corning).

For most donors, we recovered >200,000 cells of each cell type post sorting. However, in some samples, cell numbers were lower (minimum numbers of 32,000 and 48,000 for CD8^+^ T cells, 30,000 and 90,000 for CD4^+^ T cells, and 86,000 and 91,000 for macrophages, lung, and tumor samples, respectively).

After sorting, the macrophage, CD4^+^, and CD8^+^ T cell pellets were lysed in 350 µL of Buffer RLT (Qiagen, Venlo, The Netherlands) and stored at −80 °C. RNA isolation was performed following the RNeasy Micro kit protocol (Qiagen). RNA integrity was assessed using the Agilent RNA 6000 Nano kit on a 2100 Bioanalyzer (Agilent Technologies, Santa Clara, CA, USA). Isolated RNA had RNA integrity number (RIN) values >9 in most samples (mean: 9.2, min.: 6.9 in macrophages, mean: 9.6, min.: 9.0 in T cell samples), and the quantity in the majority of samples was >80 ng total RNA (mean: 640 ng, min.: 91 ng in macrophages, mean: 121 ng, min.: 30.2 ng in T cell samples). All macrophage samples, CD4^+^ T cell samples from 5 patients, and CD8^+^ T cell samples from 7 samples contained >80 ng of total RNA of good quality in both tumor and healthy tissue samples and were prepared for RNA sequencing.

RNA sequencing was carried out at the Genomics Core Facility, Department of Clinical and Molecular Medicine, NTNU. Libraries were generated for 22 + 24 samples, using Lexogen SENSE mRNA-Seq library prep kit according to the manufacturers’ instructions (Lexogen GmbH, Vienna, Austria). RNA sequencing libraries were quantitated by qPCR using KAPA Library Quantification Kit (Kapa Biosystems, Inc., Wilmington MA, USA) and pooled, normalized to 2.6 pM, and subjected to clustering on two NextSeq 500 high output flow cells. Finally, single read sequencing was performed for 75 cycles on a NextSeq 500 instrument (Illumina, Inc., San Diego, CA, USA), according to the manufacturer’s instructions. Base-calling was carried out on the NextSeq 500 instrument by RTA 2.4.6. FASTQ files were generated using bcl2fastq2 Conversion Software v2.17 (Illumina, Inc., San Diego, CA, USA). FASTQ files were filtered and trimmed (fastp v0.20.0), and transcript counts were generated using quasi alignment (Salmon v1.3.0, licensed under the GNU General Public License v3.0) to the transcriptome reference sequence (Ensembl, GRCh38 release 92). Transcript sequences were imported into the R statistical software (R Core Team, R Foundation for Statistical Computing, Vienna, Austria) and aggregated to gene counts using the tximport (v1.14.0) Bioconductor package.

Gene counts were normalized and analyzed for a differential expression using the DESeq2 (v1.28.1) Bioconductor package. DESeq2 builds a generalized linear model under the assumption of negative binomial distributed values and uses the Wald statistic for significance testing.

The Gene Ontology (GO) term and pathway enrichment are widely used approaches to identify biological themes and assess whether the number of selected genes associated with a GO term or pathway is larger than expected. GO term and pathway enrichment were carried out using the Bioconductor packages: DOSE (v 3.14.0), ReactomePA (v1.32.0), and clusterProfiler: an R package for comparing biological themes among gene clusters (v3.16.1) [27], which performed over representation analysis to investigate if GO terms or pathways were enriched for genes in our input list of differentially expressed genes (DEGs). The annotations were retrieved using the Bioconductor package org.Hs.eg.db (v3.12.0) and ReactomePA. Reactome is a free database of human biological pathways available through the Reactome Web site (https://reactome.org). *p*-values were calculated by hypergeometric distribution and adjusted for multiple comparisons by the Benjamini and Hochberg method. In addition, we utilized a network clustering approach including weighted correlation network analysis (WGCNA), on the list of DEGs to identify clusters of genes having a higher correlation among themselves than other genes. WGCNA builds a network connecting genes with high correlations, and clusters of genes that are co-expressed highlight genes and gene sets that may be explored further to look for functional origins of each cluster. For WGCNA, we used the R package WGCNA (v1.69) [28]. The gene names for CD200 and CD200R were entered into the mRNA (genechip) Kaplan-Meier plotter for lung cancer (https://kmplot.com/analysis/index.php?p=service&cancer=lung, accessed on 3 July 2020) [29] to address the association of these genes with overall survival.

Statistical analyses and visualization were done in R (v4.0.2): A Language and Environment for Statistical computing (R Core Team, R Foundation for Statistical Computing, Vienna, Austria) and Bioconductor (v3.11), packages: flowCore (v2.0.1), openCyto (v2.0.0), flowWorkspace (v4.0.6), ggcyto (v1.16.0), EnhancedVolcano (v1.6.0), and ComplexHeatmap (v2.4.3). A *p*-value of <0.05 was considered to be statistically significant.

## 3. Results

### 3.1. Immune Cells in NSCLC and Healthy Tissue Express Lineage-Specific Genes and Increased Expression of TAM-Associated Genes in Macrophages from the Tumor

We isolated macrophages, CD4^+^ T cells, and CD8^+^ T cells from the tumor as well as from normal lung tissue of 11 patients diagnosed with NSCLC (Appendix A). The majority of patients was resected at relatively early-stage NSCLC. Six patients were diagnosed with adenocarcinoma and five patients were diagnosed with squamous cell carcinoma (patient details in Table 1). Subsequently, transcriptome analysis was performed by sequencing of bulk RNA from macrophages (all 11 patients), CD4^+^ T cells (5 patients), and CD8^+^ T cells (7 patients), as we were unable to isolate RNA of sufficient quality for transcriptome analysis from some of the CD4^+^ and CD8^+^ T cell samples. Principal component analysis segregated the transcriptomic profiles of TAMs and macrophages isolated from adjacent healthy tissue, which indicates systematic differences between tumor and non-tumor macrophages that are not overshadowed by the variations between donors (Figure 1A). Similar differences, albeit less pronounced, were also observed between T cells originating from the tumor and from healthy tissue. A total of 4523 genes in the macrophage samples, 880 in the CD4^+^, and 1374 genes in the CD8^+^ T cell samples were significantly, differentially expressed between samples from tumor and from healthy tissue (adjusted *p*-value < 0.05). A list of genes mentioned in this manuscript is provided in Table 2, showing the ratio between gene expression in cells from a tumor and in cells from healthy tissue for macrophages, CD4^+^ and CD8^+^ T cells.

Hallmark genes for macrophages and T cells were highly expressed in the isolated macrophages, which include CD4^+^ and CD8^+^ T cells from tumor and non-tumor tissues, verifying the expected general phenotype of these cells (Figure 1B,C).

While typical macrophage markers such as *CD68*, *FCGR3A*, *MARCO*, and *CD14* were highly expressed in TAMs as well as non-tumor macrophage samples, the expression of canonical M1, M2, and TAM markers was more diverse. Some M1 markers (*IL1A*, *IL1B*, *CD86*), M2 markers (*CCL18*, *MRC1/CD206*, *CSF1R*), and TAM markers (*HLA-DR*, *IL1RN*, *CD163*, *ITGAM*, *SIGLEC1/CD169*) were highly expressed on both tumor and non-tumor macrophages. Expression of the M1 marker *IFNG* was relatively low in macrophages from the tumor as well as the normal lung. Many genes associated with macrophage differentiation were upregulated in TAM compared to non-tumor macrophages. These genes include M1 markers *IL23A* and, to a lesser degree, *TLR2* and *IL6*, the M2 markers *CCL18* and *CD209*, and many genes that have been described as TAM markers (*NRP2*, *IL1RN*, *HIF1A*, *APOE*, *VEGFA*, *SOCS3*, *IL10*, *CCL8*, *TGFB2*, *HGF*, and *IFITM1*). Only a few macrophage genes such as *PPARG*, *MSR1/CD204* [30], and *MARCO* were downregulated in TAMs compared to non-tumor macrophages in our dataset. We then identified the top 30 DEGs in intra-tumoral immune cells compared to their counterpart in adjacent healthy tissue (Figure 2 and Appendix A).

In TAMs compared to non-tumor macrophages, *MMP12*, a protease involved in macrophage migration [31] and the regulators of lymphocyte homing and inflammation *ENPP2* [32], *MERTK* [33] and *F13A1* [34] were upregulated. In intra-tumoral CD4^+^ and CD8^+^ T cells, genes related to regulatory T cell functions such as *MAGEH1*, a marker of intratumoral CD4^+^ effector Tregs with superior suppressive activity, *IRF4* (MUM1) [35], and *ENTPD1* were upregulated (Appendix A). Thus, our results confirmed the higher expression of various TAM markers, and several genes involved in immunosuppression in macrophages isolated from tumor tissue of NSCLC patients, than in non-tumor macrophages.

### 3.2. Genes in the Reactome Pathway «Immunoregulatory Interactions between Lymphoid and Non-Lymphoid Cells» Are Highly Regulated in NSCLC Macrophages, CD4^+^ and CD8^+^ T Cells

Bulk analysis of our entire dataset (gene expression in immune cells from tumor versus normal lung) revealed enrichment for the disease ontology terms ‘lung disease’ and ‘non-small cell lung carcinoma’, demonstrating authenticity of the data. GO terms related to the extracellular matrix, leukocyte migration, and transmembrane signaling receptor activity were enriched in the total dataset (Appendix A). GO term enrichment analysis that focused on macrophage samples only, revealed overrepresentation of genes in 46 GO terms in the subontologies: ‘molecular function,’ ‘biological process,’ and ‘cellular component’ (Appendix A). These GO terms included gene sets related to cytokine receptor binding, an extracellular structure, matrix organization, cell-cell adhesion, and the cell membrane. Extracellular matrix-organization and matrix-degradation are considered important processes in the exclusion of tumor-infiltrating lymphocytes (TILs) from the TME [36].

DEGs from macrophages, CD4^+^ and CD8^+^ T cells (differential expression in tumor versus healthy tissue) were further assessed with Reactome pathway enrichment analysis (Figure 3A). Macrophage DEGs were overrepresented in interleukin-4, interleukin-10, and interleukin-13 signaling pathway genes while CD4^+^ T cell DEGs were enriched in the Reactome pathway genes for immunosuppressive ‘interleukin-10 signaling’ and a ‘cellular response to heat stress.’ The CD8^+^ T cell DEGs were significantly overrepresented in genes involved in the ‘complement cascade.’ We identified only two Reactome pathway gene sets in which DEGs from all three investigated immune cell subsets were significantly enriched: ‘Immunoregulatory interactions between a lymphoid and a non-lymphoid cell’ and ‘cell surface interactions at the vascular wall’ (Figure 3B), the former being on top of the list of enriched pathways for both CD4^+^ and CD8^+^ T cells when sorted by an adjusted *p*-value. ‘Immunoregulatory interactions between a lymphoid and a non-lymphoid cell’ is a gene set from the Reactome database (R-HSA-198933) and includes several receptors and cell adhesion molecules thought to play a key role in modifying the response of lymphoid cells to tumor antigens in humans. Thus, the genes in this reactome pathway were selected for further analyses to identify immunosuppressive processes activated in NSCLC tumors in our data.

### 3.3. Network Clustering Analysis Identifies Networks of Genes with Highly Correlated Expression Profiles

To further identify biological networks and genes that might have a common functional origin, we utilized a network clustering approach to identify genes that are connected by high correlations. Weighted correlation network analysis (WGCNA) resulted in the identification of six gene clusters (modules) that showed correlations in gene expression patterns between macrophages, CD4^+^ and CD8^+^ T cells originating from tumor and non-tumor tissue. The modules were termed ‘blue,’ ‘yellow,’ ‘red,’ ‘green,’ ‘brown,’ and ‘turquoise’ with 101, 104, 70, 319, 40, and 160 genes, respectively (Appendix A), that showed similarities in gene regulation patterns. In the green module, many genes were upregulated in macrophage samples from the tumor, while the same cluster of genes was partly up-regulated or down-regulated in CD4 and CD8^+^ T cells from the tumor. Individual gene expression analysis of all 319 correlated genes of the ‘green’ module revealed distinct expression differences between macrophages and CD4^+^ and CD8^+^ T cells and in gene regulation between tumor and non-tumor cells (Appendix A). Looking at the list of the top 50 DEGs in the ‘green’ module (bulk analysis of all cell types), we found several genes related to known immunosuppressive pathways, such as *ADORA2B*, *IL-10*, *CD39/ENTPD1*, and *FOXP3* (Appendix A). Interestingly, the chemokine (C-X-C motif) ligand 13 (*CXCL13*), which is also known as a ‘B lymphocyte chemoattractant,’ was highly upregulated in the tumor compared to normal lung in both macrophages, CD4^+^ and CD8^+^ T cells. The top upregulated ‘green’ module gene in macrophages from the tumor was Glutamate receptor-interacting protein 1 (*GRIP1*), which was linked to the regulation of T cell activation in a previous study [37]. Similarly, the nicotinic receptor subunit *CHNRA5* was highly upregulated in macrophages and CD4^+^ T cells in tumors. Suppression of the immune response by signaling via nicotine receptors may impact immune surveillance [38]. The Notch signaling pathway has been recognized as a regulator of immune cell functions in the TME [39]. In our study, *NOTCH3* was highly upregulated in macrophages. IL-10 is regarded as a major immunosuppressive, pro-tumoral cytokine, and was highly upregulated in macrophages and CD8^+^ T cells in the tumor [40]. Thus, the genes in the ‘green’ module that are connected by correlated expression patterns and may have a common functional origin, include a plethora of dysregulated genes with proposed or well-known immunosuppressive effects in the TME.

The genes in all six clustered gene modules were further analyzed by Reactome pathway over-representation analysis. Similar to individual and un-clustered total DEG analysis in macrophages, CD4^+^ and CD8^+^ T cells, and the ‘green’ module was enriched for genes in the Reactome pathway ‘Immunoregulatory interactions between a Lymphoid and a non-Lymphoid cell’. The ´green´ module was also enriched for genes in the Reactome pathway ‘Class A/1 (Rhodopsin-like receptors)’ (Appendix A). Rhodopsin-like receptors are a large group of G-protein coupled receptors and include the chemokine receptors [41]. The Reactome pathways ‘Class A/1 (Rhodopsin-like receptors)’ and ‘Immunoregulatory interactions between a Lymphoid and a non-Lymphoid cell’ include a number of genes encoding cell adhesion molecules, chemokines, and receptors that are important in modifying the response of lymphoid cells to the tumor, which highlights these genes for further evaluation.

### 3.4. CD200/CD200R Pathway and Inhibitory Leukocyte Immunoglobulin-Like Receptors (LILRs) Represent Possible Mechanisms for Macrophage-Induced Immunosuppression/Innate Checkpoints

Next, we investigated which individual genes from the ‘Immunoregulatory interactions between a Lymphoid and a non-Lymphoid cell’ and ‘Class A/1 (Rhodopsin-like receptors)’ gene sets were most regulated in TAMs compared to macrophages from a normal lung (Appendix A). We focused on genes with a high log2 fold change (LFC) and low adjusted *p*-value (upper right corner of both plots). To aid the selection of individual genes of interest for further analysis, we also generated gene network plots for the genes assigned to the first mentioned Reactome database gene set (Figure 4A). In network plots, genes are linked by their biological functions. From these analyses, several genes were deemed of particular interest and analyzed in more detail, including *CD200R1*, *CXCL13*, *CXCR5*, *LILRB5*, and *SLAMF6* (Figure 4B).

The immunoglobulin (Ig) superfamily member CD200 receptor (*CD200R1*) is a gene listed in the gene set ‘Immunoregulatory interactions between a lymphoid and a non-lymphoid cell.’ CD200R1 is a transmembrane glycoprotein expressed mainly on the surface of myeloid cells, including macrophages. It interacts with the transmembrane glycoprotein CD200, which can be expressed on a variety of cells, including CD4^+^ and CD8^+^ T cells and tumor cells. CD200/CD200R signaling has been linked to the regulation of macrophage activation and appears to inhibit T cell responses [42]. *CD200R1* was highly upregulated on TAMs compared to macrophages from normal lung (LFC = 4.0, *p* < 0.0001) and, to a lesser degree, on CD8^+^ T cells (LFC = 1.65, *p*.adj = 0.01). In contrast, the expression of the ligand *CD200* was higher in CD4^+^ T cells in the tumor than in CD4^+^ T cells from a normal lung (LFC = 3.9, *p*.adj = 0.0009) (Figure 4B). *CD200* was also identified in the ‘green’ module of genes with correlated expression patterns and possibly common functional origin. We also looked at the association of *CD200* and *CD200R* levels with lung cancer survival using Kaplan-Meier Plotter [29]. High *CD200* expression was associated with decreased survival in that dataset (Appendix A).

By further exploring expression patterns in the gene sets ‘Immunoregulatory interactions between a Lymphoid and a non-Lymphoid cell’ and ‘Class A/1 (Rhodopsin-like receptors)’, we identified another family of transmembrane glycoprotein receptors that showed expression differences: the LILRs A and B, which are thought to regulate immune responses via their effect on myeloid cell functions [43]. *LILRB4* and *LILRB5* were upregulated in TAMs compared to macrophages from lung tissue (LFC = 1.4 and 2.0, *p*.adj < 0.01 and <0.0001) (Figure 4B). Members of the ANGPTL protein family have been proposed as LILR ligands, and *ANGPTL4* may act as a tumor suppressor or promoter of cancer metastasis, depending on the cell type and stage of cancer [44]. Expression of *ANGPTL4* was higher in TAMs than in normal tissue macrophages (LFC = 2.4, *p*.adj < 0.0001). Taken together, our data suggest an extensive activation of the CD200/CD200R and LILRB signaling pathways in immune cells of the TME in NSCLC.

Furthermore, we found highly significantly upregulation of several *SLAMF* (Signaling Lymphocytic Activation Molecule Family Member) immunoreceptors and the chemokine ligand *CXCL13* and its receptor *CXCR5* in TAMs (Figure 4B and Appendix A). In particular, *SLAMF6* and *SLAMF1* expression in TAMs was increased by LFC = 2.1 and 3.5, both *p*.adj < 0.0001, and *CXCL13* (LFC: 6.4, *p*.adj < 0.0001) and *CXCR5* (LFC: 1.8, *p*.adj = 0.001) were highly upregulated in macrophages from the tumor compared to those from a normal lung.

Several genes from the Reactome pathway ‘Class A/1 (Rhodopsin-like receptors) are related to the ATP/adenosine signaling pathway, considered to be a major immunosuppressive mechanism in the TME [45]. In our data, multiple receptors and other molecules in this pathway were highly regulated in tumor-infiltrating immune cells, including the adenosine receptors *ADORA2B* (downregulated in CD8^+^ TILs), *ADORA3* and *ADORA2A* (highly upregulated in TAMs), and the ectoenzymes *CD39* (upregulated in all tumor-infiltrating immune cell subsets) and *CD73* (highly upregulated in TAMs). CD39 and CD73 are important players in the TME, as these ectonucleotidases convert extracellular ATP to adenosine, which is an important immunosuppressive factor in the TME [45].

### 3.5. Immune Checkpoint Expression in TAMs and T Cells in NSCLC

Immunosuppressive or immunostimulatory molecules on macrophages/TAMs can influence T cells/TILs and vice versa. Several genes related to immunosuppression were detected in the top 20 DEGs from intra-tumoral immune cells as well as GO term and network analysis. This indicates an immunosuppressive phenotype of immune cells in the tumor, and a possible cross-talk between TAMs and TILs. Knowledge of this interplay may identify important immune checkpoints in NSCLC that may be targeted by existing or future immunotherapy. A summary of expression levels of several important immune checkpoint molecules in macrophages, CD4^+^ and CD8^+^ T cells, are shown in Figure 5. In our data, expression of PD-1 (*PDCD1*) was higher in CD8^+^ T cells in the tumor than in a normal lung, and PD-L1 (*CD274*) and PD-L2 (*PDCD1LG2*) were highly expressed in macrophages both from a normal lung and tumor. PD-L2 was downregulated in CD8^+^ T cells and upregulated in CD4^+^ T cells (*p*.adj = 0.003 and 0.0002). The well-known immune checkpoint *CTLA4* was upregulated in macrophages (LFC: 2.5, *p*.adj = 0.002) but not in CD4^+^ or CD8^+^ T cells. Several other genes for immune checkpoint receptor and ligand pairs were upregulated, including CD70-CD27, CD137L-CD137, OX40L-OX40, HVEM-BTLA, and B7RP1-ICOS.

## 4. Discussion

In the present study, we analyzed the gene transcripts of presorted macrophages, CD4^+^ and CD8^+^ T cells from tumor and from adjacent normal lung with RNA-sequencing, which is followed by GO term enrichment and WGCNA, to identify mediators of immunosuppression in the TME in NSCLC. By exploring the DEGs in a correlated cluster of genes (the ‘green’ module), and in two highly enriched GO terms, we identified several immunosuppressive molecules and pathways that may be activated in TAMs in NSCLC. We found a consistent differential expression of genes involved in interactions between myeloid cells and lymphocytes. Importantly, we report novel data on immune cell expression of the newly described CD200/CD200R1 pathway, and LILRs, which may be innate immune checkpoints dampening the antitumor T cell immune response in NSCLC. Our study highlights many genes with known or proposed immunosuppressive properties that are dysregulated in the TME in NSCLC, suggesting that TAMs may be an important mediator of tumor-induced immunosuppression, immune escape, and a promising target for immunotherapy.

We identified *CD200R1* as one of the genes that was highly upregulated on TAMs and intra-tumoral CD8^+^ T cells. In contrast, expression of *CD200* was higher in intra-tumoral CD4^+^ T cells than in their normal tissue counterparts. The CD200/CD200R pathway has been described as an innate immune checkpoint, and, being part of the immunoglobulin superfamily, the molecules share similar structures with other checkpoint molecules, such as PD1/PD-L1 and CTLA4-B7 [46]. Signaling in the CD200/CD200R-pathway has been thought mainly to regulate the functions of myeloid lineages of cells, and indirectly contribute to T cell response regulation. CD200/CD200R signaling may be a target for checkpoint blockade in haematological and solid tumors [47]. Previous reports on the expression of CD200/CD200R on tumor-infiltrating immune cells in NSCLC are lacking. In one study on patients with NSCLC, CD200/CD200R1 expression in tumor and stroma was investigated with IHC, and high CD200R1 expression was associated with worse survival [48]. This is in line with our data, showing correlations between expression of CD200 and CD200R and prognosis in lung cancer.

Similarly, we found higher expression of *LILRB4* and *LILRB5* in TAMs than in macrophages from normal lung tissue. LILRB represent a class of immune checkpoint molecules with similar immunosuppressive functions to that of PD-1 and CTLA-4 [18]. LILRB-ligand interaction disrupts phagocytosis clearance. LILRBs are expressed in a variety of immune cells, including macrophages and subsets of NK and T cells [14], and are reported to have tumor-promoting functions in cancer cells [18], and to mediate immunosuppression by inhibiting NK and T cell responses [18,49]. LILRB expression in cancer has been associated with enhanced tumor growth and poor patient outcomes [43], thus, suggesting blockade of LILRBs as a compelling target for immune checkpoint therapy [18,43,47,50]. While HLA class I molecules have been reported as ligands for some of the LILRBs, LILRB4 and LILRB5 remain orphan receptors [43]. Previous reports on the expression of LILRBs on immune cells in NSCLC are limited [51]. Some studies have detected mRNA expression in NSCLC cell lines and tumor samples by IHC [52,53].

SLAM family immunoreceptors are expressed on the cell surface of a variety of immune cells. They are involved in the regulation of adaptive and innate immune responses, and recently identified as a potential target for immunotherapy [54]. Similarly, a role for the CRCX5-CXCL13 axis in the TME in NSCLC has been suggested based on mouse studies and tissue and tumor samples from patients, and this signaling pathway has been implicated in the promotion of lung tumor progression [55]. These molecules and immunoreceptors might play important roles in regulating cell-cell interactions and lymphocyte infiltration [55], tumor growth, progression, and metastasis in the tumor microenvironment [56]. CXCL13 has also been reported as a possible biomarker that responds to targeted PD1/PD-L1 therapy. The highly significant upregulation of *CRCX5*, *CXCL13* on all immune cells and *SLAMF1* and *SLAMF6* in TAMs found in our study confirms these cells as the origin of these molecules in NSCLC TME, and is suggestive of a role of these molecules in the composition of an immunosuppressive TME in NSCLC.

While some purinergic receptors and adenosine receptor *ADORA2B* were downregulated in CD8^+^ and CD4^+^ T cells, TAMs showed highly upregulated *ADORA3*, *ADORA2A*, and *CD73*. In line with our previous report, *CD39* was highly expressed by both CD4^+^ and CD8^+^ T cells, as well as TAMs, derived from tumor tissue [15]. The effect of extracellular ATP signaling via purinergic receptors on tumor-infiltrating immune cells is complex, and seems to depend on the immune cell subset, type of purinergic receptor, and level of ATP/adenosine [9]. The ectonucleotidases CD39 and CD73 promote the accumulation of adenosine from ATP, having strong immunosuppressive effects in the TME [9,57]. The high expression of *CD73* and *CD39* on TAMs and T cells, and adenosine receptors *ADORA2A* and *ADORA3* on TAMs found in the present study is in line with the proposed role of the ATP/Adenosine pathway and its receptors and enzymes on T cells and TAMs as potential targets for immunotherapy [45]. The diverse M1 and M2 marker expression in the tumor as well as non-tumor macrophages [30] possibly reflects the notion that TAM phenotypes are more complex than previously believed and may not be categorized into binary states [58].

The data presented in this study is an explorative research approach to study the transcriptomic profiles of immune cells in the tumor microenvironment of NSCLC and to compare the profiles to their counterparts in healthy lung tissue. Our study has several limitations. First, the patient number in this study was too low to allow investigations of clinical traits, such as changes in expression profiles of key immunosuppressive genes during the disease course. Most of our samples were resected at relatively early NSCLC stages. While processes of immune escape evolve continuously during the stages of NSCLC [59], our findings may mainly reflect the immune cell status in the tumor microenvironment at early, rather than late or metastatic stages, where immunotherapy is usually applied. In addition, heterogeneity in tumor evolution and immune-evasion processes have been described between adenocarcinoma and squamous cell carcinoma during early stages of carcinogenesis [60,61]. The low number of immune cell transcriptome datasets in our study did not allow us to conclude on the differences in immunoregulatory processes between adenocarcinoma and squamous cell carcinoma patients. Furthermore, we were unable to isolate RNA of sufficient quality for transcriptome analysis from some CD4^+^ and CD8^+^ T cell patient samples. Hence, the statistical power in this study is lower in detecting important genes expressed by T cells in the TME, and particularly in CD4^+^ T cells. Importantly, further studies should compare immune cell gene expression profiles at different NSCLC stages before and after relapse as well as at metastatic stages. We utilized RNA sequencing on bulk-sorted immune cells, allowing control of the number of immune cells, which resulted in good quality RNA, and a predominantly homogeneous expression signature for key genes. Another popular approach is single-cell RNA sequencing, which allows discrimination of subsets within the CD4^+^ or CD8^+^ T cells or macrophages, possibly at the cost of a higher threshold for detecting important genes. We included both patients with adenocarcinomas and squamous cell carcinoma in the present study, and it is conceivable that some of the immunosuppressive mechanisms in the TME are specific for the tumor subtype. A higher number of patients would allow stratifying for tumor type and stage.

## 5. Conclusions

In conclusion, the GO term and correlation network analysis revealed multiple immunosuppressive pathways involved in signaling from TAMs in the TME. In particular, we found indications of highly activated CD200R1/CD200 signaling between TAMs and T cells in the TME and upregulation of LILRBs expression on TAMs. These two signaling pathways may be important in tumor-induced immunosuppression, and are potential targets of immunotherapy, which warrants further studies. In conjunction with our data showing multiple dysregulated genes serving as putative checkpoints controlling the anti-tumor immune response, our results highlight several possible mechanisms for immunosuppression and targets for immunotherapy in NSCLC. The present study underscores the message that we still do not know which immunosuppressive signaling pathways are the most important in the individual patient, or even in NSCLC. This insight argues for further studies by investigating the relation between immune checkpoints beyond PD-1 and CTLA-4, and prognosis. Comprehensive immune profiling of the TME in individual patients may give an ‘immunosuppression signature’ in the individual patient, which may possibly form the basis for individualized immunotherapy in NSCLC.

## Figures and Tables

**Figure 1 cancers-13-01788-f001:**
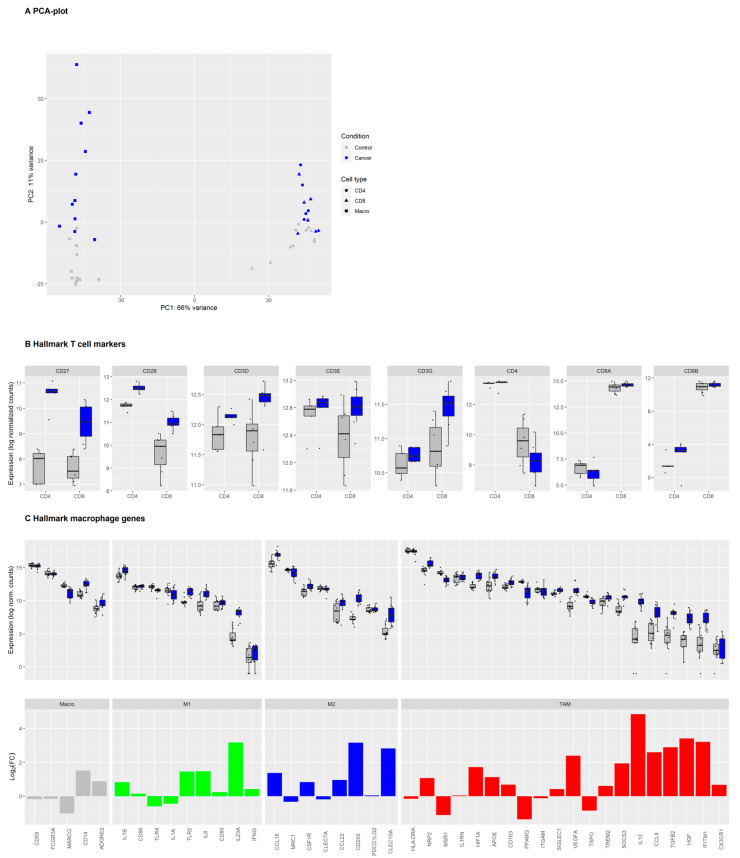
Expression of hallmark T cell and macrophage markers in tumor and normal lung immune cells in non-small-cell lung carcinoma (NSCLC). Legend: (**A**) Principal component analysis (PCA) of the log2-transformed, normalized gene expression count data of macrophages, CD4^+^ and CD8^+^ T cells originating from tumor (blue symbols) or adjacent healthy tissues (grey symbols). (**B**) Gene expression levels for individual samples of hallmark markers of CD4^+^ and CD8^+^ T cells in sorted CD4^+^ (*n* = 5) and CD8^+^ (*n* = 7) T cells from healthy tissue (grey) or tumor (blue). Boxes and whiskers display the inter-quartile range and the upper and lower limits of 1.5 x the inter-quartile range. Data beyond the end of the whiskers are outliers and plotted as points. (**C**) Upper panel: Macrophage hallmark marker gene expression in macrophages (*n* = 11) from healthy tissue (grey) or tumor (blue). Genes were grouped into general macrophage markers, canonical M1 and M2 markers, and tumor-associated macrophages (TAM) markers. Lower panel: Barplot of log2-fold change in differential gene expression levels in TAMs versus non-tumor macrophages.

**Figure 2 cancers-13-01788-f002:**
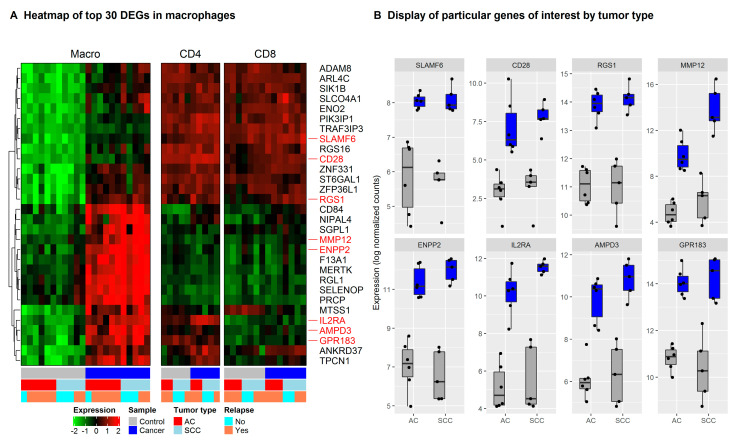
Top 30 differentially expressed genes (DEGs) in macrophages in NSCLC compared to lung tissue. Legend: (**A**) The lists of DEGs from macrophages (NSCLC versus lung tissue) were sorted for adjusted *p*-values and displayed in heatmaps. Expression of the top 30 macrophage DEGs is shown in macrophages (left), CD4^+^ (middle), and CD8^+^ T cells (right). Names of genes of particular interest are highlighted in a red color. The internal order of genes (rows) in the heatmap is disrupted by hierarchical clustering of the genes to group genes with a similar pattern of expression in each immune cell subset. The color code below the heatmaps indicates the origin of the immune cell samples. Data was sorted into samples from normal lung tissue (grey) or cancer (blue) origin and into adenocarcinoma (AC) or squamous cell carcinoma (SCC) patients. Some patients indicated experiencing a relapse within 24 months after surgery. (**B**) Boxplots showing expression profiles for individual genes of interest in macrophages isolated from healthy tissue (grey) and lung cancer (blue), by tumor type.

**Figure 3 cancers-13-01788-f003:**
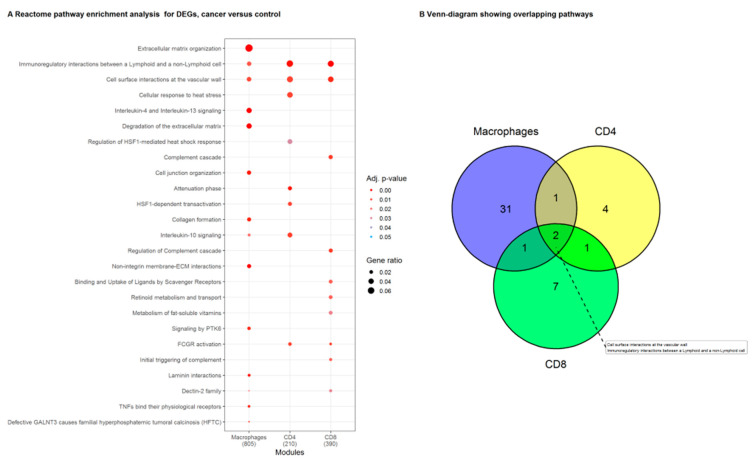
Reactome pathway enrichment analysis in macrophages, CD4^+^ and CD8^+^ T cells, from the tumor compared to their counterparts isolated from normal lung tissue. (**A**) List representing the Reactome pathways that were significantly enriched for DEGs in macrophage, CD4^+^ and CD8^+^ T cell samples. Symbol size visualizes the ratio of genes identified in our data to the genes present in the gene set. The color indicates the adjusted *p*-value. (**B**) Reactome pathways that were enriched in DEGs from the individual immune cell types (macrophages, CD4^+^ and CD8^+^ T cells). The pathways ‘Immunoregulatory interactions between a Lymphoid and a non-Lymphoid cell’, and ‘cell surface interactions at the vascular wall’ were significantly enriched in lists of DEGs from all three immune cell subsets.

**Figure 4 cancers-13-01788-f004:**
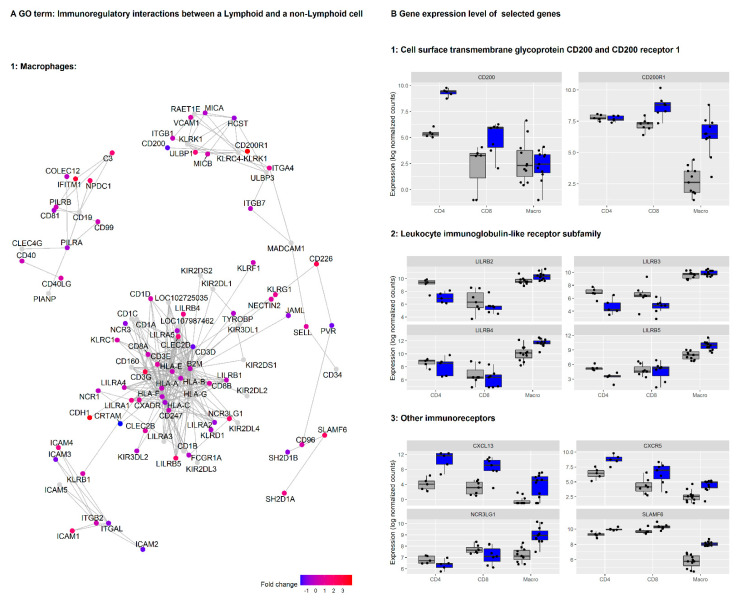
Cell surface transmembrane glycoprotein CD200 and its receptor CD200R1, and members of the Leukocyte immunoglobulin-like receptor subfamily (LILRs) are dysregulated in the tumor microenvironment (TME) in NSCLC. Legend: (**A**) the gene-network plot shows expression (log2 fold change) of genes in the GO term ‘Immunoregulatory interactions between a lymphoid and a nonlymphoid cell’ for macrophages, CD4^+^ and CD8^+^ T cells. In a gene-network plot (cnet plot in the Bioconductor package clusterProfiler), the genes are interconnected based on linkages of genes in networks based on biological functions. (**B**) 1: CD200 and its receptor CD200R1, and 2: members of the LILRs are DE in macrophages, CD4^+^ and CD8^+^ T cells from tumor compared to lung tissue. 3: Gene expression profiles of other immunoreceptors are displayed. These ligand/receptors may represent important pathways for immunosuppressive signals from TAMs to CD4^+^ or CD8^+^ T cells in the TME.

**Figure 5 cancers-13-01788-f005:**
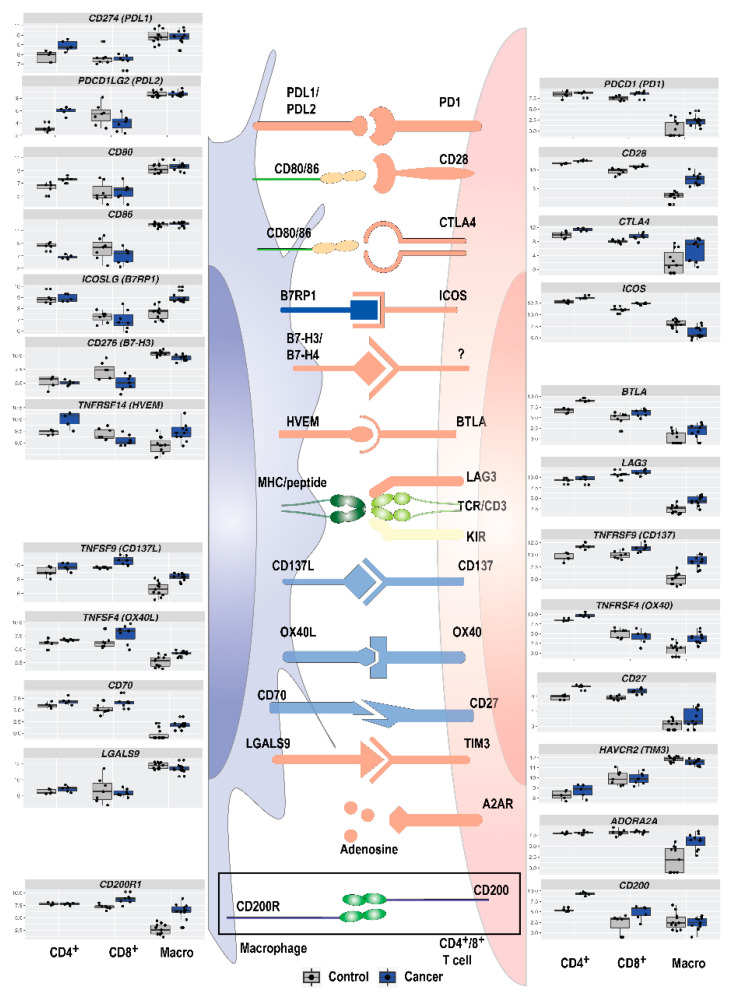
In addition to well-known immune checkpoints PD-1/PD-L1 and CTLA-4, a multitude of immune checkpoints and immunosuppressive pathways regulate the adaptive immune response in NSCLC. Legend: In the immune synapse between intra-tumoral antigen-presenting cells and effector cells, an increasing number of known or putative immune checkpoints regulate the immune response. Boxplots on the sides show expression profiles for individual immune checkpoint genes in macrophages, CD4^+^ and CD8^+^ T cells isolated from healthy tissue (grey) and lung cancer (blue).

**Table 1 cancers-13-01788-t001:** Basic characteristics of the study population.

Patient	Age	Smoker	Tumor	Subtype	PDL1	pTNM	Stage	Relapse	DFS	Immunotherapy	PFS	Samples Included (Macro/CD4/CD8)
1	70–74	former	AC	solid	1	pT2bN2-3M0	IIIA	Yes	273	pembrolizumab	14	Yes/No/No
2	75–79	former	AC	acinary	0	pT2bN1M0	IIB	Yes	891			Yes/Yes/Yes
3	70–74	former	AC	micropapillary	1	pT1cN0M0	IA3	No				Yes/No/No
4	75–79	current	SCC	keratinizing SCC	15	pT3pN0M0	IIB	No				Yes/Yes/Yes
5	70–74	current	AC	acinary	30	pT1cN0M1a	IVA	Yes	127	atezolizumab	>570 *	Yes/No/Yes
6	80–84	former	SCC	non-keratinizing SCC	0	pT2aN1M0	IIB	Yes	388			Yes/No/Yes
7	70–74	former	SCC	non-keratinizing SCC	0	pT3N0M0	IIB	No				Yes/Yes/No
8	80–84	former	SCC	keratinizing SCC	0	pT2aN0M0	IB	No				Yes/No/Yes
9	70–74	former	AC	micropapillary	0	pT3N2M0	IIIB	Yes	655			Yes/Yes/Yes
10	75–79	former	SCC	keratinizing SCC	0	pT2bN0M0	IIA	Yes	176			Yes/Yes/Yes
11	55–59	current	AC	cribriform	0	pT2aN2M0	IIIA	Yes	942			Yes/No/No

AC: Adenocarcinoma. SCC: Squamous cell carcinoma. PDL1: programmed death ligand 1, percentage of positive tumor cells analyzed by immunohistochemistry. pTNM and Stadium: histopathologic TNM Classification of Malignant Tumors (8th edition). Relapse: After a minimum of 24 months observation post surgery. DFS: Disease free survival (days after surgery). PFS: Progression free survival (days after immunotherapy). *: No progression detected until the end of the study period.

**Table 2 cancers-13-01788-t002:** Ratio between gene expression in cells from non-small cell lung carcinoma tissues and from healthy tissues for macrophages and T cells.

Gene Symbol	Gene Name	Fold-Change * (Macrophages)	Fold-Change * (CD4^+^ T Cells)	Fold-Change * (CD8^+^ T Cells)
*ADORA*	Adenosine receptors	Gene family		
*AGPAT4*	Phospholipid acyltransferase	0.58	−1.47	−1.81
*ANGPTL4*	Angiopoietin-like protein family 4	2.41	#	#
*APOE*	Apolipoprotein E	1.13	#	#
*B7RP1 (CD275, ICOSLG)*	ICOS ligand	1.33	#	#
*BTLA (CD272)*	B-lymphocyte and T-lymphocyte attenuator	2.17	2.38	#
*CCL18*	Chemokine (C-C motif) ligand 18	1.38	#	#
*CCL8*	Chemokine (C-C motif) ligand 8	2.59	−2.27	#
*CHRNA5*	Neuronal acetylcholine receptor subunit alpha-5 acetylcholine receptor subunit	#	#	#
*CSF1R*	Colony stimulating factor 1 receptor	#	−1.76	#
*CTLA-4*	cytotoxic T lymphocyte protein 4	2.51	#	#
*CXCL13*	Chemokine (C-X-C motif) ligand 13	6.41	6.66	5.94
*CXCR5*	C-X-C chemokine receptor type 5	1.77	2.31	2.26
*DC-SIGN (CD209)*	Dendritic cell-specific intercellular adhesion molecule-3-Grabbing non-integrin	3.15	#	#
*ENPP2*	Ectonucleotide pyrophosphatase/phosphodiesterase family member 2	4.56	#	#
*ENTPD1 (CD39)*	Ectonucleoside triphosphate diphosphohydrolase-1	1.37	2.73	2
*F13A1*	Coagulation factor XIII A chain	5.07	#	#
*FCGR3A (CD16a)*	Low affinity immunoglobulin gamma Fc region receptor III-A	#	−1.77	−1.9
*FOXP3*	Forkhead box P3	3.23	2.79	1.92
*GRIP1*	Glutamate receptor-interacting protein 1	7.98	#	#
*HGF*	Hepatocyte growth factor	3.41	#	#
*HIF1A*	Hypoxia-inducible factor 1-alpha	1.72	#	#
*HLA-DR*	HLA class II histocompatibility antigen, DR alpha chain	#	#	−1.21
*HVEM (TNFRSF14)*	Tumor necrosis factor receptor superfamily member 14	0.66	#	#
*ICOS (CD278)*	Inducible T-cell co-stimulator	−1.54	#	1.29
*IFITM1*	Interferon-induced transmembrane protein 1	3.21	#	#
*IFNG*	Interferon gamma	#	#	1.03
*IL10*	Interleukin 10	4.83	#	1.99
*IL1A*	Interleukin 1 alpha	#	#	#
*IL1B*	Interleukin 1 beta	#	−1.95	#
*IL1RN*	Interleukin-1 receptor antagonist	#	#	−1.38
*IL1RN*	Interleukin 1 Receptor Antagonist	#	#	−1.38
*IL23A*	Interleukin-23 subunit alpha	3.16	#	#
*IL2RA (CD25)*	Interleukin-2 receptor alpha chain	5.1	1.99	#
*IL6*	Interleukin 6	1.47	#	#
*IRF4 (MUM1)*	Interferon regulatory factor 4	2.48	#	#
*ITGAM (CD11B, Mac-1)*	Integrin alpha M	#	−2.19	−2.08
*LILRA*	Leukocyte immunoglobulin-like receptor subfamily A	Gene family		
*LILRB*	Leukocyte immunoglobulin-like receptor subfamily B	Gene family		
*MAGEH1*	Melanoma-associated antigen H1	0.96	3.42	#
*MARCO*	Macrophage receptor with collagenous structure	−1.01	−2.44	−2.85
*MERTK*	Proto-oncogene tyrosine-protein kinase MER	-	#	#
*MMP12*	Matrix metalloproteinase-12	7.79	#	#
*MRC1 (CD206)*	Mannose receptor C-type 1	#	#	−2.28
*MSR1 (CD204)*	Macrophage scavenger receptor 1	−1.10	#	−2.29
*NOTCH3*	Neurogenic locus notch homolog protein 3	4.21	#	4.3
*NRP2*	Neuropilin 2	1.07	#	−1.47
*NT5E (CD73)*	5′-nucleotidase	3.26	#	#
*OX40 (CD134, TNFRSF4)*	Tumor necrosis factor receptor superfamily, member 4	2.98	#	#
*OX40L (CD252, TNFSF4)*	Tumor necrosis factor (ligand) superfamily, member 4	1.37	#	1.62
*PDCD1 (PD-1)*	programmed cell death protein 1	1.4	#	#
*PDCD1LG2*	Programmed cell death 1 ligand 2	#	2.85	−1.81
*PD-L1 (CD274)*	programmed cell death protein ligand 1	#	#	#
*PPARG*	Peroxisome proliferator-activated receptor gamma	−1.36	#	−2.3
*SIGLEC1 (CD169)*	Sialo-adhesin	#	#	−2.01
*SLAMF*	signaling lymphocytic activation molecule family	Gene family		
*SOCS3*	Suppressor of cytokine signaling 3	1.94	#	#
*TGFB2*	Transforming growth factor-beta 2	2.89	#	#
*TLR2*	Toll-like receptor 2	1.45	#	#
*VEGFA*	Vascular endothelial growth factor A	2.39	#	#

Gene symbols and gene names for genes mentioned in this manuscript, and the ratio between gene expression in cells from tumor and in cells from healthy tissue for macrophages, CD4^+^, and CD8^+^ T cells. *: Comparison of immune cells from tumor versus immune cells from normal lung; #: Not significant.

## Data Availability

The data presented in this study are available on request from the corresponding author.

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
