# Peer review of "Analysis of Intra-Tumoral Macrophages and T Cells in Non-Small Cell Lung Cancer (NSCLC) Indicates a Role for Immune Checkpoint and CD200-CD200R Interactions"

_cancers, 2021, doi:10.3390/cancers13081788_

Round 1

Reviewer 1 Report

In this article, the authors investigated the gene expression profiles of intratumoral immune cells from NSCLC patients and compared them to the expression profiles of their counterparts in normal adjacent lung; particularly, the authors focused on the immune-regulatory role of TAMs and their pathways of interaction with TILs.  The article is interesting and focuses the scenario on new potential immunological target of innate immunity. The research has several important limitations to be underlined, thus in order to be considered for publishing a series of major revisions are needed, as follows:

  • Firstly, analyses were performed in resected early-stage NSCLCs. This model cannot represent an ideal model for understanding resistance pathways for immunotherapy in advanced/metastatic lung cancer, where immunotherapy is usually applied in clinical practice. Indeed, several studies suggested significant change in TME and immune-infiltration moving from early stage to metastatic stage, as result of the dynamic process of carcinogenesis. Additionally, TME heterogeneity can also be found by analyzing primary tumors and different metastatic sites. In this cohort of patients, 40% of patients experienced disease recurrence, but any clinical or survival information (DFS, OS, PFS to immunotherapy) has been provided by the authors and related to laboratory findings. Despite the few number of patients evaluated, this data should be important to understand the translational value of the pre-clinical findings. This findings should be reported and addressed as severe limitations of the performed work.
  • Moreover, different immune-evasion events have been described during early carcinogenesis between lung adenocarcinomas and squamous cell carcinomas, focusing on more peculiar role of TME in SCC early immune escaping (MC Granahan et al. Cell 2017). For these reasons, the authors should provide difference in TME finding between the two subtypes.
  • Lastly, the researcher failed to obtain complete lymphocyte transcriptome analysis from half of patients; consequently, technical challenges and procedural limitation should be discussed in the study and incomplete result of lymphocytes analysis should not be presented.
  • Article should be present as explorative research and technical limitation should be widely discussed
  • Remove the incomplete data of lymphocytes analysis
  • Stratification according to tumor histology and clinical correlation should be presented
  • Limitation to apply these findings in advanced/metastatic setting, where usually immunotherapy is applied, should be extensively discussed in conclusions.

Author Response

To both reviewer 1 and 2.

Reviewer 2 Report

  1. The authors are advised to validate a part of the results with immunohistochemistry of NSCLC tissue samples.
  2. The authors are advised to refer to the below article. The intratumoral distribution influences the prognostic impact of CD68- and CD204-positive macrophages in non-small cell lung cancer. Li Z, Maeda D, Yoshida M, Umakoshi M, Nanjo H, Shiraishi K, Saito M, Kohno T, Konno H, Saito H, Minamiya Y, Goto A. Lung Cancer. 2018 Sep;123:127-135. doi: 10.1016/j.lungcan.2018.07.015.

Author Response

To both reviewers 1 and 2.

Round 2

Reviewer 2 Report

The quality of the article has been substantially improved in response to the reviewers comments.